# An Examination of Ability Emotional Intelligence and Its Relationships with Fluid and Crystallized Abilities in a Student Sample

**DOI:** 10.3390/jintelligence8020018

**Published:** 2020-04-24

**Authors:** Juliane Völker

**Affiliations:** Department of Psychology, University of Trier, D-54296 Trier, Germany; voelker@uni-trier.de

**Keywords:** emotional intelligence (EI), fluid abilities (Gf), crystallized abilities (Gc), Intelligence Structure Battery (INSBAT), Geneva Emotional Competence Test (GECo)

## Abstract

This study investigated correlative, factorial, and structural relationships between scores for ability emotional intelligence in the workplace (measured with the Geneva Emotional Competence Test), as well as fluid and crystallized abilities (measured with the Intelligence Structure Battery), carried out by a 188-participant student sample. Confirming existing research, recognition, understanding, and management of emotions were related primarily to crystallized ability tests measuring general knowledge, verbal fluency, and knowledge of word meaning. Meanwhile, emotion regulation was the least correlated with any other cognitive or emotional ability. In line with research on the trainability of emotional intelligence, these results may support the notion that emotional abilities are subject to acquired knowledge, where situational (i.e., workplace-specific) emotional intelligence may depend on accumulating relevant experiences.

## 1. Introduction

Doing small talk about job entrance experiences, I recall hearing the sentiment rather often that education did prepare young people for the job, but it didn’t really prepare them for dealing with the people there (which was probably intended to sound more or less humorous in at least some instances). Indeed, the idea that the organizational settings are also a space for emotions is relatively novel ([59]). One major contribution to an emotional wakeup call certainly came from the surfacing of the construct called Emotional Intelligence (EI). Today, there exist three main streams in EI research (ability, trait, and mixed-model perspectives; [2]). Ability EI views emotional intelligence from the standard intelligence perspective, focusing on objective performance measures of the construct. Trait EI encompasses emotional personality dispositions and people’s subjective perceptions thereof ([43]). Mixed-model EI constitutes the broadest approach, including behavioral markers such as capabilities, skills, and personal characteristics ([57]). The differences of these models entail varied approaches to measurement and applications, which bears challenges in terms of validity and conceptual understanding of EI in general ([37]; [56]); however, it also offers multifaceted perspectives on interacting emotional capabilities and dispositions. Adopting the cognitive ability perspective, the present article focuses on the ability conception of EI.

Back in the 1990s, Mayer and Salovey proposed the concept of EI as a unique ability to reason about emotions. Their four-branch model became the commonly accepted model of ability EI ([31]; [47]): “Perceiving emotions” addresses the correct identification of emotions and emotional information; “facilitating thoughts using emotion” means utilizing emotions to assist intellectual processing; “understanding emotions” refers to the use of emotional knowledge to label affective states and recognize complex relationships between them; and “managing emotions” means the ability to consciously regulate one’s own, as well as others’ emotions (for more detail about the skills related to each branch, see [32]). The perception and facilitation branches are sometimes referred to as lower-order branches, or experiential EI, whereas understanding and management of emotions are conceived as higher-order, strategic EI ([15]). The test employed most often to measure the four branches is the Mayer-Salovey-Caruso Emotional Intelligence Test (MSCEIT).

The initial and continual appeal of EI was that it supposedly benefits various aspects of daily life. For instance, EI’s promise of positive impact on the workplace has roused plenty of research, interventions, and commercial interest (for reviews, see [59]; [20]). It has also roused advances in the measurement of ability EI beyond the MSCEIT’s conceptual and psychometric limitations ([16]; [46]). Tests corresponding specifically to workplace contexts have recently been developed with the North Dakota Emotional Abilities Test (NEAT; [23]), or the Geneva Emotional Competence Test (GECo; [48]).

Meanwhile, researchers have discussed whether EI is indeed to be considered a form of intelligence, which would entail that measures of ability EI need to be associated with standard intelligence. In fact, Legree and colleagues found that MSCEIT scores loaded significantly onto the general factor *g* ([25]; [24]). But is EI actually a new type of intelligence, or just a reflection of standard intelligence applied to emotional contexts? To test this, one approach is to investigate the status of ability EI as a second-stratum factor in the Cattel-Horn-Carroll (CHC) model of cognitive abilities ([33]). CHC theory is currently the most widely accepted theory of mental abilities, as it manages the feat of integrating a diverse and complex range of human abilities into a comprehensive framework by grouping narrow abilities (first stratum) to broader cognitive domains (second stratum), which all fall under a general intelligence factor *g* (third stratum; the positive correlation of ability variables is known as Spearman’s law of positive manifold; [55]). From this perspective, EI defines a set of interrelated abilities that are distinct enough from other cognitive abilities to form a unique domain, but also broadly related to other ability domains to be considered a part of the general intelligence factor *g* ([51]). Arguably, the most important contribution in this regard was provided by [28] ([28]), who conducted an exhaustive structural analysis with multiple cognitive ability tests and the MSCEIT in a sample of 688 students. By evaluating the fit of several possible structure models to their obtained data, they concluded that EI is best understood as a broad ability factor worth implementing in a revision of the CHC model. Very recently, [11] ([11]) replicated these findings with several other emotional ability measures tapping into the field of EI.

Consideration of connections between EI and other domains, usually fluid and crystallized abilities, has also been subject to investigation. This concerns fluid and crystallized ability’s conceptual links to EI as well as the statistical relationships among measures for these ability domains. From the perspective of standard cognitive intelligence, fluid abilities (Gf) comprise capabilities to discern relationships between stimuli, to understand implications and confer logical conclusions. Gf abilities are most in demand when solving abstract or novel problems. Crystallized abilities (Gc, also “comprehension-knowledge” in CHC taxonomy) refer to the depth of acquired knowledge as well as verbal fluency and understanding of words and concepts ([51]). On a conceptual level, [15] ([15]) proposed to distinguish between a fluid component of EI which addresses emotion processing, and a crystallized component which is based on emotional knowledge (see also [7]; [13]). This is strengthened by relationships between the four EI branches and Gf and Gc tests. Associations between EI and Gf were found to be moderate ([14]), connections between EI and Gc were equivalent in some ([38]), or even stronger in other studies ([12]; [21]; [27]). In meta-analyses, weak to modest relationships between the four emotional ability branches and Gf and Gc were found ([38]; [44]). It is noteworthy that the strongest associations with Gc surfaced for tests of emotion understanding ([38]; [3]).

[52] ([52]) suggested an even more detailed interdependent conceptualization of the four-branch model with other cognitive functions besides Gf and Gc. For example, perceiving emotions may draw from visual or auditory processing abilities when one detects emotion in the facial expression and tonality of the voice of another person. Facilitating thoughts using emotions may influence knowledge-related and fluid reasoning abilities, which concerns utilization of affective states influencing attention and perception, in a given context. In turn, fluid and crystallized abilities could exert effects on understanding emotions: To learn about emotions is to process emotional information, to reason about them, and to build emotional knowledge. Finally, emotion management as a higher-order ability would not only draw from the aforementioned lower-order emotional abilities, but could also depend on cognitive processing speed and working memory capacity. In sum, while EI abilities appear to form a separate group of mental abilities, they are also intricately related to other cognitive functions, with each branch and associated skills depending on each other and related broad abilities.

### 1.1. EI Trainability—Evidence for a Knowledge Base?

EI’s relationship with knowledge-based abilities is especially interesting from the perspective of EI trainings. (In educational practice, the similar concept of “emotional literacy” has been used to foster socio-emotional learning in schoolchildren, e.g., [40]; [58]). Such trainings appeared to be effective by producing stable improvements in EI, especially when based on the ability model ([18]; [54]). In particular, and matching the correlative findings mentioned above, emotion understanding seems to benefit from imparting explicit emotion knowledge as a foundation to develop emotionally competent behavior. Acquiring knowledge about emotions is one aspect; however, actually applying this knowledge in everyday life is yet another ([14]; [18]).

How should this gap between “knowing” and “doing” be approached? There are trainings targeting professional contexts that gear more towards the mixed-model approach of EI, broadening the concept to include skills such as self-awareness or interpersonal sensitivity ([8]; [17]). Such trainings aim to teach behavior strategies in line with EI, and these too appear to be effective. A meta-analysis by [30] ([30]) reported medium sized effects in support of the trainability of EI both grounded on the four-branch ability model or the mixed-model approach based on skills. Unfortunately, detailed insights into trainings were not provided by the referenced publications—if the trainings were effective by expanding emotional knowledge (in line with ability EI) or otherwise (e.g., workshops and discussions, adhering to mixed-model EI) remained inconclusive.

However, accumulating emotion knowledge and skills is not just a matter of explicit trainings spanning a couple of interventions, but mostly day-to-day experience over the entire course of one’s life. [34] ([34]) have presumed that emotion understanding and emotion management follow a path similar to knowledge-related abilities: they increase and stabilize until early adulthood, and then slowly decrease with age. Evidence is provided by [5] ([5]): abilities to accurately perceive emotions, facilitate thought, and manage emotions reached a peak in adult samples and decreased approaching older age, while emotion understanding declined linearly from adolescence onwards. This could reflect the general decline of cognitive functioning with age, however, having a higher educational background appears to act as a protective factor against an age-related decrease of EI ([36]).

How would ability EI, fluid and crystallized abilities be related to each other in a young student sample, given the developmental evidence from earlier studies? A novel perspective to add to this picture was the examination not of general EI by using the MSCEIT, but of specific emotional abilities by using a measure that has been developed for workplace contexts; namely, the GECo. The GECo differs from the MSCEIT in terms of theory-driven development of stimuli and responses (compared to the MSCEIT’s consensus and expert scoring to determine correct responses, which is one of its major points of criticism for causing response patterns to be skewed towards higher performances; [16]; [48]), as well as in its situational task design and multiple-choice format scoring (compared to MSCEIT’s rating scores). Therefore, GECo scores seem more fit to predict performance indicators (such as grades) than the MSCEIT does, and they reflect acquired procedural knowledge subjected to work experience ([48]). University students may be relatively unacquainted with tasks presenting workplace contexts, and therefore the present investigation was expected to provide an additional, possibly more differentiated insight into the connections between context-specific emotional and fluid as well as crystallized abilities.

### 1.2. The Present Study

The objective was to investigate the correlations and factor structure between workplace ability EI, fluid and crystallized abilities in a sample of university students. First, it was expected to find weak to moderate correlations between test scores as evidence for positive manifold across the three domains. Using component analysis and structural equation modeling, results should replicate EI, Gf and Gc as three distinct group factors amongst test scores. The latent factors, in turn, were expected to be interrelated as parts of the general factor *g*.

Furthermore, the scores were tested for effects of sex, age and education level. Although the sample was expected to be quite homogenous, differences between sexes and education levels are nevertheless to be reckoned with, in ways that males may score higher on fluid ability tests ([39]), females may score higher on emotional ability tests ([44]; [5]), and higher education levels being associated with better test performances ([36]). A younger sample is also likely to achieve higher scores on fluid rather than crystallized abilities, given that crystallized and emotional abilities were found to cultivate with age ([5]). Whether emotional abilities are differently related to fluid (lower-order, experiential EI) or crystallized abilities (higher-order, strategic EI) was to be explored ([38]).

## 2. Materials and Methods

### 2.1. Sample

A sample of 188 participants (163 females) completed tests assessing emotional and cognitive abilities (age range 18 to 46, *M* = 21.86, *SD* = 4.09). All participants were students at the University of Trier, Germany. The majority were in their first or second semester at university (*n* = 100). Most participants graduated from secondary school with a university entrance qualification (*n* = 172), some already had a university degree (*n* = 12), and four graduated with a general certificate of secondary education. Male and female participants were distributed evenly across education levels, χ^2^(2) = 2.08, *p* = 0.354 (Table 1), and did not differ in age, *F* = 2.41, *p* = 0.123. Not surprisingly, age was associated with education level, *r_SP_* = 0.234, *p* < 0.001.

### 2.2. Tests

Emotional abilities were assessed with the German version of the Geneva Emotional Competence Test (GECo; [48]). The GECo provides scores for emotion recognition, emotion understanding, emotion regulation and emotion management. Dissimilar to the four-branch model, the test drops facilitation because of conceptual and empirical redundancies ([16]; [46]). Furthermore, emotion management addresses behavior strategies to manage other people’s emotions, whereas the use of cognitive strategies to manage own negative emotions (emotion regulation) is assessed separately.

Emotion recognition tasks present short video clips showing actors who express an emotion in a fictional language. The correct emotion must then be selected from several possible answers (see also the short form of the Geneva Emotion Recognition Test, GERT-S; [49]). Tests of emotion understanding, emotion regulation and emotion management employ situational tasks describing workplace contexts. Emotion understanding assesses how well one understands causes and consequences of one’s own and other people’s emotions by correctly appraising brief scenarios. For example, if someone has to give a presentation in a foreign language they are having trouble with, which emotion are they most likely to feel? Emotion regulation presents tasks in which a behavior to deal with own negative emotional situations is to be selected from multiple possible answers, each representing a regulation strategy that is either adaptive (e.g., reappraisal) or maladaptive (e.g., rumination). Participants choose a response based on the reaction they would most likely show. High performance on emotion regulation is defined by a consistent use of adaptive strategies. Emotion management refers to behaviors in response to someone else’s usually negative emotions. Responses were designed to represent different conflict management strategies (e.g., compromise or avoidance), varying in their situational appropriateness depending on situational factors such as social pressure or organizational norms. Again, participants are required to indicate the behavior they would typically show in the presented scenarios. For the interested reader, more detail on the underlying theories and development of each subtest is provided in [48] ([48]).

Fluid and crystallized abilities were assessed with the Intelligence Structure Battery (INSBAT; [1]). The INSBAT is designed to measure six ability factors in accordance with the CHC model, but its modular form allows for testing ability domains selectively. Subtests for Gf comprised numerical inductive reasoning, with tasks requiring the logical completion of number sequences; figural inductive reasoning, consisting of 3 × 3 figure matrices where the correct missing piece needs to be identified from multiple possible answers; and verbal deductive reasoning, requiring the test taker to select a correct conclusion drawn from two given statements. The subtests for Gc were general knowledge, which requires the completion of definitions; verbal fluency, where nouns must be identified from a sequence of unordered letters; and word meaning, which asks for a correct synonym for each presented word from multiple possible answers. The tests are adaptive regarding sociodemographic parameters and task performance of the test taker. Resulting scores for each group of subtests were obtained and used for subsequent analyses.

### 2.3. Procedure

The GECo was administered online and took approximately 60 min to complete. Afterwards, participants were invited to take part in an additional laboratory assessment of cognitive abilities. There, the Gf and Gc modules from the INSBAT were administered using the digital testing software Vienna Test System ([53]). The tests were conducted in separate cubicles equipped with 15.6-inch HD screen laptops. The two modules took approximately 87 min to complete, including a 5-min break between the Gf and Gc module.

The assessments were carried out in accordance with the Declaration of Helsinki and the recommendations of the Ethics Committee of the University of Trier. Prior to both assessments, participants were informed that they could terminate participation at any time and for undisclosed reasons. All data was anonymized and scores from the separate tests were matched using individual codes. All participants granted their consent and received course credit for their participation.

## 3. Results

Analyses were run in SPSS 26.0.0.0. All participants’ data was used. The GECo scores of ten participants could not be matched to INSBAT scores due to erroneous codes. The missing cases were excluded listwise.

### 3.1. Descriptive Statistics and Correlations

Descriptive statistics and bivariate correlations between emotional and cognitive abilities scores are presented in Table A1 (see Appendix A). Most Gf and Gc test scores were correlated at the *p* < 0.01 level, demonstrating positive manifold. Fewer in number, several significant correlations occurred amongst EI scores (*p*s < 0.01). Emotion recognition, understanding, and management showed significant correlations with some cognitive test scores (*p*s < 0.05–0.01), while emotion regulation showed a negative correlation with verbal fluency (*p* < 0.05). Thus, with an exception of the negative correlation, scores for emotional, fluid, and crystallized abilities were mostly related as expected.

### 3.2. Effects of Age, Sex and Education Level

Two cognitive test scores correlated significantly with age: figural inductive reasoning (*r* = 0.15, *p* < 0.05) and verbal fluency (*r* = 0.21, *p* < 0.01). For emotional abilities, a significant correlation between age and emotion regulation (*r* = 0.15, *p* < 0.05) was observed.

An analysis of variance (ANOVA) revealed several significant differences between male and female participants. More specifically, males obtained a higher score on numerical inductive reasoning (*M* = 105.00) than females (*M* = 98.59), *F*(1, 186) = 5.31, *p* = 0.022, η^2^ = 0.028, and on figural inductive reasoning males scored higher (*M* = 105.28) than females as well (*M* = 98.03), *F*(1, 186) = 6.47, *p* = 0.012, η^2^ = 0.034. For EI, a marginally significant difference between sexes was found on emotion management, *F*(1, 177) = 3.63, *p* = 0.059, η^2^ = 0.020. Females attained a slightly higher score (*M* = 0.53) than males (*M* = 0.48).

When age and education level were controlled for, the differences regarding numerical inductive reasoning, *F*(1, 184) = 5.33, *p* = 0.022, η^2^ = 0.028, and figural inductive reasoning, *F*(1, 184) = 5.42, *p* = 0.021, η^2^ = 0.029, remained significant. Therefore, the better performance of male participants could not be explained by age and educational achievement. In contrast, the difference regarding emotion management became even more significant after controlling for age and education level, *F*(1, 174) = 4.82, *p* = 0.029, η^2^ = 0.027.

### 3.3. Principal Component Analysis (PCA)

To examine the latent structure underlying the test scores, a PCA was conducted on the ten variables measuring EI, Gf and Gc. Barlett’s test of sphericity suggested that a suitable portion of correlations among variables is present for the analysis to yield reliable results, χ^2^(45) = 232.88, *p* < 0.001. On top of that, the Kaiser-Meyer-Olkin criterion (KMO) and Measures of sampling adequacy (MSA) assessed whether appropriate proportions of correlations relative to partial correlations were present (indicated by values approaching one). The KMO of 0.727 implied that an acceptable ratio is given in the overall sample. Similarly, MSA reflect the ratio of correlations to partial correlations for each variable in the set, which helps gauge its respective appropriateness for inclusion in the analysis. The MSA for emotion regulation (0.463) was below the recommended cut-off of 0.50. This means that emotion regulation caused diffuse correlations, and finding a reliable estimate of common variance accountable for latent factors appeared to be limited. Removing emotion regulation from the analyses slightly improved overall adequacy (KMO = 0.751), however, this did not gravely change the results, but reduced the proportions of explained variance. Therefore, the subsequent PCA included emotion regulation despite its impaired adequacy.

Four components extracting variance portions approaching or larger than one were found, together explaining 60.38% of variance. Because of positive manifold amongst cognitive abilities, items were rotated using an oblique procedure. In accordance with the small sample size, the Direct Oblimin method was employed. Items clustering on the same components suggested that component 1 represented Gf with all three respective test scores, while component 2 represented emotion regulation. Emotion management and verbal fluency marked component 3, and finally, component 4 represented emotion understanding, emotion recognition, general knowledge and word meaning. The pattern matrix and portions of variances covered by the components are reported in Table 2.

The loadings of the highest loading variable on each component were high on one (from 0.71 to 0.87), but not on a second component for emotional and fluid ability scores. This indicated that emotional and fluid ability variables formed a strong association with but one component, which facilitated their interpretation. This pattern was not so pronounced for crystallized abilities and emotion recognition, where loadings ranged from 0.43 to 0.59. Emotion recognition and verbal fluency also loaded onto both the third and fourth component. Moderate to high proportions of variance of the test scores were covered by the four components (from 0.45 to 0.80, with a mean of *M* = 0.61). These parameters signify that, despite the comparatively small sample size and issues with adequacy caused by emotion regulation, an acceptable amount of variance could still be interpreted in terms of the extracted components ([26]). The components were not or weakly correlated (Table 3).

### 3.4. Structure Models

Following the examples of [28] ([28]) and [11] ([11]), two a priori specified structural equation models were tested for fit to the data. Model 1, the unidimensional model, tested the hypothesis that as part of general intelligence, all subtests should load onto the latent factor *g* (Figure 1). Model 2, the theoretical group factor model, examined if the test scores would load onto their respective theoretical factor, and whether the latent factors are weakly to moderately correlated as premised by positive manifold (Figure 2). Heeding the previously reported correlations and PCA results, a third, adjusted group factor model was also tested (Model 3). This third model tested emotion recognition, emotion understanding, general knowledge, and word meaning as indicators for a latent factor (say, Gc & EI). Emotion management and verbal fluency were tested as indicators for another latent factor (say, EM & VF). Emotion regulation was omitted in this model because it would have been a single indicator for a latent factor (Figure 3). For the identification of each model, the indicators with the highest loadings found in the PCA were constrained to 1. All models were fit using SPSS AMOS 26.0.0.0 with maximum likelihood estimation. Fit indices were selected for model comparison (Akaike information criterion AIC, comparative fit index CFI,) incremental fit relative to independence models (normed-fit index NFI, Tucker-Lewis index TLI), and to gauge approximation (root mean square error of approximation RMSEA) and residuals (standardized root mean square residual SRMR) of each model. Fit indices are displayed in Table 4.

Within the figures, dashed lines indicate non-significant path coefficients. Double-headed arrows between latent factors (ellipses) display correlations. The values presented are standardized parameters. Significance levels are indicated with * *p* < 0.05, * *p* < 0.01, and *** *p* < 0.001. The indicators which were constrained for model identification are marked with ^1^. Values inside indicator variables (rectangles) show squared multiple correlations. Errors for each indicator variable are displayed as latent variables at the bottom of each model.

The unidimensional model indicated no agreeable fit, χ^2^(35) = 56.82, *p* = 0.011. All Gf and Gc scores loaded significantly onto the general factor *g*, whereas of EI scores emotion recognition, emotion understanding, and emotion management attained significance (*p*s < 0.001). Latent *g* explained a highly significant portion of variance (*p* < 0.001). The theoretical group factor model provided a better fit, χ^2^(32) = 38.60, *p* = 0.196, CFI > 0.95, RMSEA < 0.06, TLI > 0.90. All test scores for Gf and Gc loaded significantly onto their respective latent factor. Of EI scores, emotion recognition, emotion understanding and emotion management loaded significantly onto EI (*p* = 0.001). There were significant correlations between Gf and Gc (*p* < 0.001), Gc and EI (*p* < 0.001), and Gf and EI (*p* = 0.001). Latent Gf and Gc explained significant portions of variance (*p*s < 0.001), and so did latent EI (*p* = 0.008). Finally, the adjusted group factor model resulted in the best fit of the three models, χ^2^(24) = 24.34, *p* = 0.442. It obtained the lowest AIC and exceeded most indices’ recommended cutoffs (CFI > 0.95, RMSEA < 0.06, SRMR < 0.05, TLI > 0.90, and NFI ≈ 0.90,). All path coefficients and correlations attained significance.

## 4. Discussion

The present study examined emotional, fluid and crystallized abilities in a young and mostly female student sample. The obtained fluid and crystallized test scores centered mostly on the population norm of 100 and within the standard deviation of 15, with fluid abilities scoring slightly higher than crystallized abilities as was expected due to the young mean age of the sample. For the GECo, no reliable norms are given as of now, yet the scores seem comparable to those reported in Schlegel and Mortillaro (studies 1, 2, and 5 in [48]; mind, however, that the test was still undergoing development with impact on the scoring). It was found that Gf emerged most clearly as a distinct broad ability, while the pattern turned out more diffuse for EI and Gc. EI and Gc were interrelated in some ways, yet distinct in others. Gf and Gc were positively associated as has been found in multiple studies before. Meanwhile, EI related more to Gc than Gf. Furthermore, the plainest indicator of EI appeared to be emotion recognition. Recognizing and understanding of emotions also related the most clearly to fluid and crystallized abilities. The possible inferences about the nature of developing higher-order, situational EI abilities are discussed and reviewed with regard to sample characteristics.

### 4.1. Relations between Emotional, Fluid and Crystallized Tasks

Positive manifold was observed, yet correlations were in part smaller and fewer in number than expected. Correlations amongst emotional ability scores occurred between emotion recognition and emotion understanding, indicating their developmental convergence ([9]). The ability to recognize emotions was also correlated with better emotion management. In comparison, in Schlegel and Mortillaro’s studies, emotional ability scores correlated with each other generally at the *p* < 0.01 level in a student and university member sample, whose mean age was about ten years older than the present sample (M = 32.2 years, study 5 in [48]). This is not surprising, as [22] ([22]) argued that emotional and cognitive abilities converge with age, so that relationships between them turn out smaller when examining students compared to older samples. Regarding relationships with other broad abilities, similar although stronger relationships between EI and Gc are reported by several other authors ([48]; [12]; [39]; [45]), in some cases showing that these relationships increased with age ([50]). In fact, the present results resemble those reported by [12] ([12]). Their sample characteristics were most similar to the present one (see further below), and they reported an association of EI with Gc, but lack thereof with Gf. Presently, though EI was associated with Gf, relations were stronger with Gc.

Regarding broad abilities, the three latent group factors and their interrelations could be replicated, complying with the findings of [28] ([28]) and [11] ([11]). Considering that the adjusted group factor model (Model 3) was extremely sample specific, the improvements in fit compared to the theoretical group factor model could be negligible. Also, the variance explanations of Gc became diffused when split up to cluster with separate EI factors. Therefore, it seems reasonable to evaluate the theoretical group factor model (Model 2) as the more meaningful model.

Summed up, these correlative findings as well as the explorations of the factor structure in PCA and SEM could indicate that developing EI indeed draws from crystallized abilities. It may only be in prospect that with increasing experience emotional abilities develop stronger statistical associations with each other to form distinct ability EI.

### 4.2. The Special Case of Emotion Regulation

What is puzzling is that emotion regulation was an isolated ability. It was neither related to emotion perception, understanding, or management, nor to crystallized abilities. It was even negatively related to fluid abilities. Where does the ability to regulate one’s own emotions come from, if it is indeed different from management of other people’s emotions ([9])? If it were knowledge-related, it may be uncorrelated because of the situational task type and thus an effect of the lack of workplace experience as assumed above. But if this were true, wouldn’t emotion regulation be closer related to fluid abilities, as participants were supposedly dealing with novel problems? Perhaps, thinking back to [52]’s ([52]) model of integrating emotional and cognitive abilities, emotion regulation may be a unique ability that draws from cognitive processes other than Gf and Gc. According to [14] ([14]), one feature of emotion regulation is that it involves selective attention to emotional stimuli and choice of strategies to attain regulatory goals. [9] ([9]) also discussed emotion attention regulation as an extension of the ability EI model, contemplating the (dis)engagement processes involved in attention to emotional stimuli ([10]). It may be possible that regulation of attention to (internal) emotional information moderates regulatory processes involving these emotions. For instance, [6] ([6]) found that in an Iowa Gambling Task, high EI exerted a positive influence on cognitive control (which is associated with directing attentional processes) when faced with disadvantageous outcomes. Beyond the consideration of abilities, adaptive emotion regulation is also subject to behavior dispositions anchored in personality. It has been demonstrated that trait EI—self-perceptions concerning emotional dispositions and behavior ([43])—as well as the Big Five personality dimensions predicted emotion regulation and coping styles ([41]). Specifically, a higher trait EI was associated with differentiated employment of emotion regulation styles in order to reduce negative emotions while maintaining positive ones ([35]; [42]). Possibly, emotion regulation from the perspective of ability EI mediates the effect of personality (such as trait EI and neuroticism) on successful coping by promoting task-oriented approaches instead of emotion-focused strategies ([4]). It was only recently that scholars attempted to review the predating emotion regulation research within the field of EI ([21]; [34]; [41]; [19]). Investigating emotion regulation in light of cognitive processes aside from emotional, fluid or crystallized abilities or associated personality domains appears to be a highly interesting prospect to pursue in future studies.

### 4.3. Impact of Sample Characteristics

Finally, the influence of the specific sample needs to be addressed. The results presented fit existing literature; however, because the sample was homogenous in terms of demographic factors, the following inferences are made very cautiously.

Students performed slightly better on fluid than on crystallized ability tests and thus confirming the expectation regarding age effects. Age was correlated with one Gf subtest (figural inductive reasoning), one Gc subtest (verbal fluency), and with one EI ability (emotion regulation), but not with any other scores. [12] ([12]) also reported no correlations between age and MSCEIT scores, and their sample was similarly aged (*M* = 22.5 years, also university students with a similar proportion of males and females). The low mean of age ([34]), but, considering EI, also the workplace specificity of the GECo might be responsible for this finding.

There were performance differences between males and females on fluid ability tests. A similar result was also reported by [39] ([39]), where men performed better than women in reasoning and processing abilities, among others. Interestingly, differences in age and education level could not explain the present effects. Why males scored higher in some tasks needs to be explained. In contrast, females outshone males in emotion management, which was also found by [12] ([12]) and [27] ([27]). The idea that females are traditionally more socialized to display emotional competencies could explain why they attain higher scores. Possibly, emotional knowledge is treated much differently by society from general or other types of knowledge. Males and females may also be differentially motivated to acquire emotional knowledge ([27]). One may even argue if crystallized abilities underlying emotional management in the workplace would be better understood as domain-specific knowledge (Gkn), a tentatively identified ability domain within CHC theory ([33]). However, considering the omnipresence of emotions in life, one would expect it to be general rather than specific knowledge, although this may be liable to cultural differences. Along with emotion regulation, there seem to be more conceptual and empirical interrelations to be explored than just fluid and crystallized abilities.

### 4.4. Limitations

Assessments of abilities were rather selective with three task types for each Gf and Gc, four for EI. With measures for just two cognitive abilities there is only limited reliability for estimating the general factor *g*. Concerning the GECo, a substantial part of the tasks was presented in a situational design. As [38] ([38]) observed, the response format could have inflated the association found with Gc (compared to studies that employed the MSCEIT). Although [27] ([27]) results have not shown that the response format significantly impacted the factor structure, the issue for the present study should not be discarded if not directly contrasted with other EI tests. Using multiple measures for ability EI and related abilities could paint a clearer picture, for instance, by comparing MSCEIT and GECo performances controlled for workplace experience.

Regarding analysis methods, larger sample sizes and more indicators for latent variables are needed to render component analysis and structure equation modelling more reliable. Despite high loadings, agreeable portions of variance covered by the components, and good to excellent fit indices, the ability structure found in this study should be tested for both stability and generalizability.

Finally, to control for possible moderators, a more diverse and balanced sample regarding age, sexes, education levels, and even different cultural backgrounds would be more insightful into how and in whom structures of latent ability factors are developed.

## 5. Conclusions

Overall, the correlational and structural results from this study fit the theoretical and empirical literature on the relationships between emotional, fluid, and crystallized abilities, while also providing further evidence for their unique pattern in a young student sample in their early university education. Ultimately, to investigate the development of workplace ability EI, its dependence on crystallized abilities, training of EI skills and relation to age, longitudinal studies should be employed ([36]). Future studies may test if the factor structure of EI and related abilities is malleable by EI trainings or acquired experience, and if these truly work by constructing knowledge-related levels of ability EI. From [18]’s ([18]) meta-analysis, workshop approaches seem to be particularly promising. Also, [29] ([29]) provide recommendations for fostering competencies related to ability EI in higher education; for example, by course design and assignments. However, determining which interventions match the conceptualization of ability EI in association with Gc to develop relevant emotion skills is a grand endeavor yet to be systematically addressed. Subsequently, examining possible convergent courses of development in line with other models of EI (trait and mixed-model) for fostering emotional self-esteem and competent interpersonal behavior in the face of each other’s emotions might be promising, as well ([57]). For EI in general, this could emphasize the importance of its many facets in everyday life, and assess utility of each model in the contexts it is most relevant (e.g., cognitive performance and ability EI, or mixed-model EI’s value in organizational contexts; [56]). This may not only help researchers to better understand the development process of emotional abilities (or “emotional literacy”), but it may also benefit young people to understand the developmental path to emotional functionality in their future workplaces. In any field, knowledge-based abilities indicate expertise, and expertise requires experience, even (or maybe especially) if they concern emotional abilities in the working world.

## Figures and Tables

**Figure 1 jintelligence-08-00018-f001:**
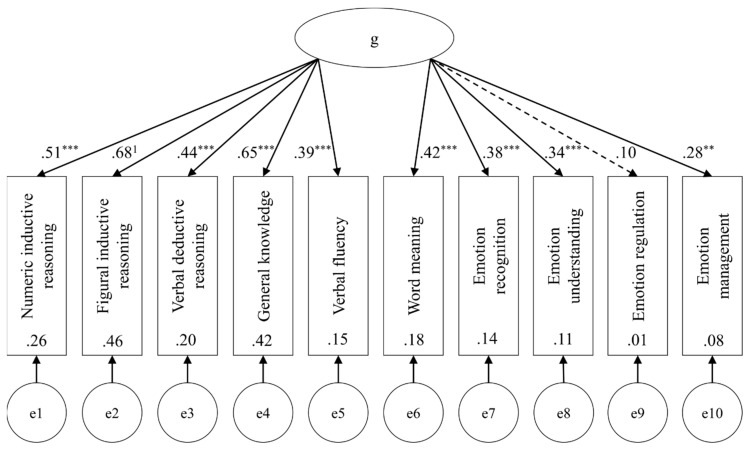
Unidimensional model (Model 1).

**Figure 2 jintelligence-08-00018-f002:**
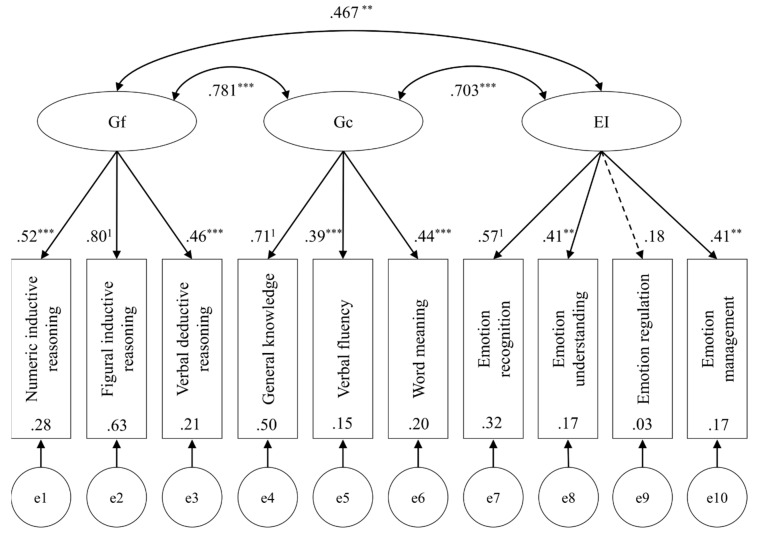
Theoretical group factor model (Model 2).

**Figure 3 jintelligence-08-00018-f003:**
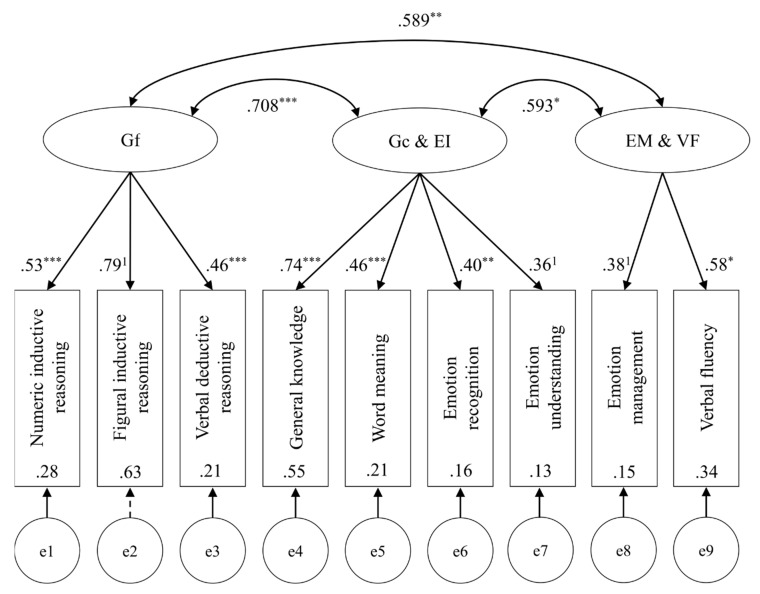
Adjusted group factor model (Model 3).

**Table 1 jintelligence-08-00018-t001:** Education levels of males and females.

	*N*	Education Level ^1^		
		1	2	3
Male	25	1	21	3
Female	163	3	151	9
Total	188	4	172	12

^1^ 1 = general secondary degree; 2 = university entrance; 3 = university degree.

**Table 2 jintelligence-08-00018-t002:** Pattern matrix ^1^ and variances covered by components.

	Components				Proportion of Variance Covered by Components
	1	2	3	4	
Emotion recognition	−0.13	−0.01	0.45	−0.59	0.58
Emotion understanding	0.00	0.17	−0.10	−**0.78**	0.62
Emotion regulation	0.22	**0.87**	0.19	−0.01	0.80
Emotion management	−0.02	0.16	**0.87**	0.00	0.77
Numerical inductive reasoning	0.66	−0.01	−0.10	−0.11	0.48
Figural inductive reasoning	**0.79**	−0.04	0.09	−0.01	0.66
Verbal deductive reasoning	**0.71**	0.15	−0.02	0.07	0.48
General knowledge	0.38	−0.09	0.19	−0.43	0.52
Verbal fluency	0.28	−0.55	0.49	0.14	0.69
Word meaning	0.24	−0.24	−0.11	−0.51	0.45
Extracted variance covered by component	2.73	1.23	1.08	0.998	

^1^ Rotation method: Direct Oblimin. Loadings >±0.70 are displayed in boldface.

**Table 3 jintelligence-08-00018-t003:** Component correlations.

	2	3	4
1	−0.12	0.18	−0.27
2		−0.03	0.04
3			0.02

**Table 4 jintelligence-08-00018-t004:** Fit indices for the three structure models.

	AIC	CFI	RMSEA	SRMR	NFI	TLI
Model 1	116.818	0.887	0.059	0.061	0.762	0.855
Model 2	104.600	0.966	0.034	0.051	0.838	0.952
Model 3	84.811	0.996	0.013	0.043	0.896	0.992

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
