# Peer review of "An Examination of Ability Emotional Intelligence and Its Relationships with Fluid and Crystallized Abilities in a Student Sample"

_jintelligence, 2020, doi:10.3390/jintelligence8020018_

Round 1

Reviewer 1 Report

Overall, I found this to be fascinating and I like the approach taken.  One suggestion is to not only cover the MSCEIT theoretical background/abilities approach, but to explain the other approaches with a little detail and cite those, a well.  I especially think that this overall approach to the intersection of concepts would be very interesting to replicate with the behavioral (mixed-model) perspective on EI. 

I also would love to see a bit more on the explanation/theory on the CHC model and how EI is usually found as distinct from cognitive models, but you are making a connection it falls under g and within the CHC model. I see you cited that MacCann et al study, but for those unfamiliar it could use a bit more support.  Is that the only study that found that connection?  How could that be integrated with the other frameworks of EI?

Overall, more theoretical support and explanation would be useful, in terms of the combining of emotional and cognitive measures in the analysis. 

The of EI is also discussed prior to methods, but as this is a single data collection period- that isn't really relevant to this method-- perhaps this would be better suited for the discussion and implications/future direction? 

Author Response

Thank you very much for the feedback and the useful comments!

Point 1: One suggestion is to not only cover the MSCEIT theoretical background/abilities approach, but to explain the other approaches with a little detail and cite those, a well.  

Response 1: I briefly explained the three models (ability, trait, mixed) in the beginning paragraph (lines 26-30), and picked up on the differentiation in the conclusion (lines 445-450).

Point 2: I also would love to see a bit more on the explanation/theory on the CHC model and how EI is usually found as distinct from cognitive models, but you are making a connection it falls under g and within the CHC model. I see you cited that MacCann et al study, but for those unfamiliar it could use a bit more support.  Is that the only study that found that connection?  How could that be integrated with the other frameworks of EI? Overall, more theoretical support and explanation would be useful, in terms of the combining of emotional and cognitive measures in the analysis.

Response 2: I added a little more explanation about CHC theory (lines 51-55) and for analysing ability EI in line with CHC and g, I have included research of Legree et al. (2016, 2014) as further evidence (lines 52-62). Indeed, MacCann et al (2014) and Evans et al. (2019) appear to be the only studies conducting SEM in line with CHC theory. Methodological differences between GECo and MSCEIT are now remarked in lines 133-137.

Point 3: The of EI is also discussed prior to methods, but as this is a single data collection period- that isn't really relevant to this method-- perhaps this would be better suited for the discussion and implications/future direction?

Response 3: There is a word missing in the final paragraph of your review – I would presume it concerns “trainability” or “development” of EI in section 1.1? I found it important to be explained at this point because it frames the expected results in the student sample:  If certain facets of EI are presumably not yet as developed, one would expect to find lower scores (and conceive that they would probably turn out higher in a second assessment years later, if this were actually a longitudinal study). It’s similar to the “younger age = higher Gf / older age = higher Gc” argument, and the results presented in this section also support the relationship between EI and Gc (both expected to turn out lower in a young sample, yet are correlated).

I haven’t made any changes to this section in regards to content (for now), but if there is anything that needs further clarification please feel free to point it out!

Reviewer 2 Report

This is a well-conducted empirical study of the associations among emotional intelligence (EI) and cognitive intelligence measures in a healthy sample.  The study provides worthwhile data on these associations, which are important for creating a higher-order model of the structure of ability more generally.  The authors use a state-of-the-art measure of EI, notably a relatively recent test that includes well-validated factors.

The paper is very well written.  The introduction is well researched, with the necessary background to motivate the investigation.

The findings are not surprising, but this is not a problem as much as a valuable opportunity for replication.  In particular, crystallized intelligence relates to several key EI measures and verbal intelligence to others.  The authors point out the implications of these findings namely that EI may be trainable by accumulating greater crystallized knowledge.

A larger sample than N=111 would have been preferable, but this sample is sufficient for publication still to be worthwhile.

Analyses were sophisticated, and illustrated a relatively clear set of results.

Overall, this is an excellent piece of research.  I learned from it and wish the authors the best for it and for their future work.

Author Response

Thank you very much for your feedback and encouraging comments!

Point 1: A larger sample than N=111 would have been preferable, but this sample is sufficient for publication still to be worthwhile.

Response 1: I have managed to recruit a larger sample (N = 188), with some (minor) impacts on the results.

Some other revisision concern alternative EI models (trait and mixed-model), comments on CHC theory, and methods (PCA, fit indices for SEM).